# Modeling HPV-Associated Disease and Cancer Using the Cottontail Rabbit Papillomavirus

**DOI:** 10.3390/v14091964

**Published:** 2022-09-04

**Authors:** Nancy M. Cladel, Jie Xu, Xuwen Peng, Pengfei Jiang, Neil D. Christensen, Zhi-Ming Zheng, Jiafen Hu

**Affiliations:** 1The Jake Gittlen Laboratories for Cancer Research, College of Medicine, Pennsylvania State University, Hershey, PA 17033, USA; 2Department of Pathology, Pennsylvania State University College of Medicine, Hershey, PA 17033, USA; 3Center for Advanced Models for Translational Sciences and Therapeutics, University of Michigan Medical Center, University of Michigan Medical School, Ann Arbor, MI 48109, USA; 4Department of Comparative Medicine, Pennsylvania State University College of Medicine, Hershey, PA 17033, USA; 5Institute of Molecular Virology and Immunology, Department of Microbiology & Immunology, School of Basic Medical Sciences, Wenzhou Medical University, Wenzhou 325035, China; 6Department of Microbiology and Immunology, Pennsylvania State University College of Medicine, Hershey, PA 17033, USA; 7Tumor Virus RNA Biology Section, HIV Dynamics and Replication Program, National Cancer Institute, National Institutes of Health, Frederick, MD 21702, USA

**Keywords:** rabbit, papillomavirus, CRPV, HPV, tumor regression, disease progression, cancer, gene modified rabbits, RNAseq, codon optimization, wound healing, immune responses

## Abstract

Approximately 5% of all human cancers are attributable to human papillomavirus (HPV) infections. HPV-associated diseases and cancers remain a substantial public health and economic burden worldwide despite the availability of prophylactic HPV vaccines. Current diagnosis and treatments for HPV-associated diseases and cancers are predominantly based on cell/tissue morphological examination and/or testing for the presence of high-risk HPV types. There is a lack of robust targets/markers to improve the accuracy of diagnosis and treatments. Several naturally occurring animal papillomavirus models have been established as surrogates to study HPV pathogenesis. Among them, the Cottontail rabbit papillomavirus (CRPV) model has become known as the gold standard. This model has played a pivotal role in the successful development of vaccines now available to prevent HPV infections. Over the past eighty years, the CRPV model has been widely applied to study HPV carcinogenesis. Taking advantage of a large panel of functional mutant CRPV genomes with distinct, reproducible, and predictable phenotypes, we have gained a deeper understanding of viral–host interaction during tumor progression. In recent years, the application of genome-wide RNA-seq analysis to the CRPV model has allowed us to learn and validate changes that parallel those reported in HPV-associated cancers. In addition, we have established a selection of gene-modified rabbit lines to facilitate mechanistic studies and the development of novel therapeutic strategies. In the current review, we summarize some significant findings that have advanced our understanding of HPV pathogenesis and highlight the implication of the development of novel gene-modified rabbits to future mechanistic studies.

## 1. Introduction

Human papillomavirus (HPV)-associated diseases and cancers remain a significant public health problem worldwide [1]. Due to the species-specific properties of HPV, several naturally occurring animal papillomavirus models have been critical in studying HPV pathogenesis [2,3]. Among these preclinical models, the Cottontail rabbit papillomavirus (CRPV) was the first identified papillomavirus and the CRPV model has been widely used to study viral–host interactions for HPV-associated diseases and cancers since the first report by Shope in 1933 [4,5,6,7,8]. The extensive genetic and functional homology of CRPV with high-risk HPVs has made this model system a gold standard for testing novel anti-viral and anti-tumor treatments leading to clinical applications and providing the first proof-of-evidence for the current HPV vaccines [9,10,11,12,13,14,15,16,17]. Over the past eighty years since the discovery of this tumor virus [4], especially after the CRPV genome sequence was reported in 1985 [18], we have gained a significant understanding about viral pathogenesis by using tools such as genetic modification to alter this virus genome (mutations /insertions /deletions) without destroying its ability to induce tumors. Several key milestones in CRPV studies correlating to breakthroughs in HPV cancer research are updated in Figure 1 from previous reviews [8,19,20,21,22,23,24,25,26,27,28].

CRPV has significant biosafety advantages relative to HPVs in preclinical experiments because its species specificity ensures that it does not pose harm to humans and other animals. Therefore, the CRPV model is ideal to test many novel anti-viral and anti-tumor compounds, as well as novel vaccines [9]. To facilitate vaccine development for both prophylactic and therapeutic purposes, many vaccine strategies have been developed including peptide, protein, and DNA vaccines targeting both viral early and late genes (E1, E2, E6, E7, E8, L1, L2) [29,30,31,32,33,34,35,36,37,38,39,40,41]. Some of these strategies have moved on to clinical trials (see review paper [8]). We also synthesized the HPV/CRPV pseudovirus to test novel vaccines, including a broadly protective minor capsid protein L2 vaccine in the CRPV rabbit model [42,43,44,45].

In addition to different mutant viral genomes, rabbits with different genetic backgrounds (inbred, outbred, transgenic, and gene knockout) have been used to advance our understanding of the interaction of viral pathogenesis and host immunogenicity [7,33,46,47,48,49,50,51,52,53,54]. In the current review, which is not inclusive of all of the research performed in the rabbit papillomavirus field, we focused on some of our recent findings relating to viral pathogenesis in the post-genetic modification era (Figure 1) and highlight recent advances in gene-modified rabbits [55] that can be used for future studies.

## 2. Cottontail Rabbit Papillomavirus (CRPV)-Associated Pathogenesis

The CRPV genome exhibits a genetic structure and biology similar to those of high-risk HPVs [3,8,27,56]. Three oncogenes, E6, E7, and E8 (an equivalent for E5 of HPV, which is now also called E10), corresponding to those of HPVs have been identified [33,57,58,59,60]. To investigate the oncogenicity and immunogenicity of viral genes, a large panel of mutant CRPV genomes have been generated by different groups over the years, including 300 plus mutant genomes generated in our laboratory [20,23,25,26,50,53,58,61,62,63,64]. Some unique features of our mutant CRPV genomes are summarized in Table 1.

CRPV-infected tissues can either progress to cancer, maintain persistent and benign, or regress completely depending on the viral and host genetic background (Figure 2) [8,50,53,65]. The CRPV rabbit model is an excellent model to assess the role of both early and late genes in vivo because infection can be initiated with the viral DNA cloned into a plasmid [20,23,27,50,66,67,68]. Intriguingly, some of these mutant CRPV genomes display unique phenotypes in disease outcomes at predictable time frames [8,27,49]. Using an improved pre-wounding strategy established in our laboratory, we were able to achieve consistent and reproducible results among different experiments [69].

### 2.1. Increased Viral Infection and Tumor Growth Using a Pre-Wounding Strategy

In the original study, Shope used a scarification strategy to successfully inoculate wart suspensions and to induce tumor growth on the skin of both wild and domestic rabbits [4]. This method has been adopted in most published studies for CRPV viral infections. Since the development of genetic modification technology, mutant CRPV genomes have been generated to further understand the function of individual genes in the viral life cycle and tumor progression. The best strategies to effectively induce infections with viral DNA have been a road block for researchers until the pre-wounding method was tested and validated [20,22,23,24,25,27,33,50,53,56,58,62,64,65,66,67,69,70,71,72]. The pre-wounding technique greatly improved the efficiency of infections initiated by plasmid DNA. Using this technique, plasmid loads as low as 40 ng yielded infection [69]. Interestingly, this new method also significantly increased viral infectivity by a thousand-fold, and increased induction of tumors from a dilution of 10^−2^ of viral stock to as low as 10^–5^ from the same viral stock [69]. In addition to improvements in both reproducibility and consistency, the pre-wounding technique is cost-effective considering the limited resources of viral stock and the cost of making large quantities of highly purified viral DNA plasmid [50]. It was especially helpful in increasing the sensitivity of some viral mutants, such as the E8ATGko mutant [33,58], that are less viable than the wild type. Using the pre-wounding method, we demonstrated that an E8ATGko mutant genome induced significantly smaller tumors than those of the wild type [64], whereas no lesions were found using a gene-gun delivery method by another group [58]. Therefore, this improved pre-wounding technique for viral inoculation played a significant role to gain a more accurate and deeper understanding of the in vivo oncogenicity of the individual oncogenes.

The mechanisms underlying the improved viral infection by pre-wounding in our inoculation protocol need further investigation [69]. Skin wounding triggers innate immune responses including inflammatory reactions via recruiting immune cells to counteract local infections [73,74]. We postulate that the wounding strategy plus CRPV infections further promote the local chronic inflammation that has been associated with cancer development [75,76,77]. Coincidently, we have identified a panel of wound healing-related molecules in CRPV tumors using the genome-wide transcriptome assay for which a high homology is shared between rabbits and humans [73,74,78,79,80,81]. Some of these molecules are significantly dysregulated in CRPV tumors [82]. Recent studies confirmed the important role of wound healing-related molecules including Arginase1 and Cox-2 in cutaneous wound repair; interestingly, these molecules were found to be dysregulated in CRPV-induced lesions [81,82]. Therefore, this model holds the promise of further understanding the role of these inflammation-associated molecules in HPV-associated viral infection, persistence, and tumor progression, which would improve our ability to identify interventions to treat and prevent HPV-associated diseases and cancers.

In addition to local infections, we also demonstrated that productive infections could be established by delivery of virions or viral DNA intravenously [82]. The intravenous infection was first reported in the original study by Shope, using wart suspensions [4]. Using careful controls and different viral doses, our study provided solid and new proof-of-evidence to show that papillomavirus especially viral DNA can be transmitted through the bloodstream and induce local infections at pre-wounded sites of domestic rabbits [82]. These findings suggest the possibility that the same could pertain in humans [83].

### 2.2. The Use of Mutant CRPV Genomes to Understand the Viral Life Cycle In Vivo

We have made modifications in both the early and late genes of the CRPV genome [27,33,35,36,50,71]. Among many mutant genomes that induce visible skin tumors on rabbits, the regions that tolerate insertions, deletions, and mutations cluster in the two late capsid genes, L1 and L2. As expected, most mutants with changes in the L1 and L2 genes did not significantly reduce the capacity to induce tumor growth in vivo [35,36,62,84].

#### 2.2.1. Early Genes Play a Crucial Role in Viral Life Cycle and Tumor Growth

The early genes E1, E2, E6, and E7 are essential for tumor growth in vivo [20,21,27,60]. However, we were able to insert small fragments at the end of the E6 and E7 genes of some of the mutant genomes without losing the capacity for induction of tumor growth [51,84]. Many of these early gene-modified constructs became less vigorous in promoting tumor growth even with the pre-wounding method [27]. It should be noted that viral DNA can be readily detected in the lesions of **wild cottontail rabbits** after infection with the wild-type CRPV genome cloned as a plasmid by in situ hybridization [8]. However, the tumors induced by the same genome in **New Zealand White (NZW)** rabbits show much lower copies of viral DNA despite similar levels of tumor growth and antibody detection in these animals [8,35,56,82]. These findings suggest that the plasmid may interfere in some way in the domestic laboratory rabbits, while the Cottontail rabbits could overcome this interference (unpublished observations). Plasmid interference in the laboratory rabbits is supported by our findings that the release of DNA from the vector prior to infection resulted in increased L1 signals in the resulting lesions [56]. To increase the viability of the mutant constructs, we leveraged the knowledge of naturally identified tandem repeat papillomavirus sequences [82] and generated tandem repeat CRPV constructs as described in our previous publications [26,84]. These tandem repeat mutant constructs with duplicate numbers of early (E6, E7, E1, E2) and/or late (L1 and L2) genes that showed increased infectivity resulting in tumor outgrowth in vivo [26,84] display additive pathological functions and could be also complemental [26]. The tandem repeat strategy was not only used to explore the impact of different early and late genes in the viral life cycle in vivo [27,84] but also to generate hybrid constructs with HPV genes/epitopes inserted into different regions for vaccine development against HPV-associated diseases and cancers [51,84].

#### 2.2.2. Synonymous Codon Optimization Increases Oncogenicity and Immunogenicity of the Virus

As in the case of HPVs, the CRPV genome contains many rare codons, presumably to escape host immune and miRNA surveillance by inhibiting the expression of its early and late genes [72,85,86,87]. To release the brake on this suppression, we introduced synonymous codons (without changes in the protein sequences) into the oncogenes E6 and E7 of the wild-type CRPV to match mammalian codons [25,72]. The codon-optimized E6 (CoE6) and E7 (CoE7) proteins promote cell proliferation in vitro [25,72]. CoE7 also induces primary centrosome duplication errors leading to abnormal centrosome numbers (>2) in CoE7 transfected cells (our unpublished observations), as shown in HPV16E7 [87,88]. Therefore, aneuploidy associated with codon-optimized CRPV (designated as CoCRPV) E6 and E7 may have played a role in accelerated cancer development observed for some CoCRPV genomes [25,72]. We identified one particular CoCRPV containing 15 and 18 synonymous codon changes in E6 and E7, respectively, that could induce cancers within 16 weeks post infection [25]. The accelerated cancer development by the CoCRPV genomes is characterized by a disruption of the basement membrane and invasion into the dermis as early as week 10 post infection, and contrasts significantly with the one-year average time scale to cancer for wild-type CRPV [25]. The lesions generated from the CoCRPV genomes contain higher viral copies, suggesting an increased viral replication in the codon-optimized CRPV-infected cells [25]. Comparable levels of E6 and E7 transcripts between the CoCRPV and wtCRPV lesions suggest that the levels of these two viral transcripts may not be critical in triggering a malignant transition [89,90,91], even though they might be important for tumor initiation, given that the UV light reactivation of latent CRPV infections significantly increased the E6/E7 transcripts [92]. In contrast, a third of the CoCRPV papillomas showed a greater tendency for regression or reduced growth [72]. This outcome may be due to larger amounts of oncoproteins being produced in codon-modified papillomas and subsequently targeted by the immune system, as we observed increased immune cell infiltration in these lesions [41,72]. We look forward to utilizing these unique constructs for further elucidating the functions of these oncoproteins in the viral life cycle and pathogenesis.

#### 2.2.3. Early Gene E6 Is Important for Tumor Regression

HPV E6 has been shown to play a key role in disease progression by binding and degrading tumor suppressor protein p53 [93,94,95]. Similarly, CRPV E6 has been shown to bind tumor suppressors [59]. Based on the phenotype, two CRPV viral strains have been isolated: the progressive strain and the regressive strain that mimics high- and low-risk HPV types, respectively [53,65]. By swapping the E6 genes between these two unique CRPV strains, we observed that the E6 of the progressive strain is the key oncogene for viral persistence, a prerequisite for cancer development [50,59]. To further understand E6 function in vivo, we generated several CRPV genomes with hybrid progressive and regressive E6 [27,96]. We determined that the carboxyl terminus of regressive E6 is crucial for the regressive phenotype [50]. Interestingly, the same construct could display the opposite phenotype based on the host’s genetic background [50,53], or when the host T cells were depleted, which parallels the increased HPV disease and cancers in organ transplant patients [57]. The contribution of the host immune control of papillomavirus infections has benefited from studies on tumors that regressed [97,98,99]. Tumor regression correlated with infiltration of the CD4 and CD8 T cells that target early genes, such as E2 and E6 [100,101,102,103,104,105,106,107,108]. The CRPV model will be a useful tool to gain deeper understanding of the roles of infiltrating cells in the regression, using novel technology such as single-cell omics.

The constructs with hybrid E6 between the progressive CRPV E6 and E6 of the rabbit oral papillomavirus (ROPV), a mucosotropic papillomavirus, have been used to further understand tissue tropism and the underlying mechanisms [36,96,109,110,111]. Interestingly, all of these latter hybrid constructs failed to promote tumor growth in the skin sites of rabbits [27]. Despite this lack of viability in vivo, some of these hybrid E6 constructs show oncogenicity in vitro [109], suggesting that in vivo failure may be related to the tissue specificity of CRPV versus ROPV. Further studies will be needed to better understand the role of E6 in the pathogenesis and regression of different tissues.

## 3. Genetic Analyses of Changes during CRPV Infections

By taking advantage of recent genome-wide transcriptome analyses, we have begun to gain a deeper understanding of the changes occurring at the molecular level during infection [8,82,90]. Unbiased whole-genome RNA-seq analysis has been utilized and host gene transcript profiling of tumor tissues has been reported in our recent study [82].

### 3.1. Host Changes during Viral Infection

Immune cell infiltration is correlated with CRPV-induced tumor regression, as demonstrated in other PV models and HPV [41,103,104,105,106,107,108]. The changes at the transcription level of the host genes that were identified and correlated with CRPV infection have been reported [82,90,112,113], including common signal transduction pathways/ molecules in HPV specimens. Using two representative host genes that play critical roles in DNA repair (Apobec2) and inflammation (IL36r) that were identified in the wild-type CRPV-infected tissues as examples [82], we also observed a similar expression pattern in persistent and benign tumors induced by a CRPVE8ATGko (E8m) mutant genome suggesting that both the wild type and E8 mutant interfere with these pathways [33,58]. As we have a large selection of mutant CRPV genomes with different phenotypes that can be used for comparative studies, we may determine whether the differences we observed in some of the genes/pathways among different mutant genome--induced lesions are predictive for tumor growth and disease trajectory. These can be measured by in vitro and in vivo T cell function assays [51], neutralization assays, ELISA, Western blot, immuno-precipitation, and cytokine profiling assays [8,24,82].

### 3.2. In Situ Analysis of Tissues at Different Disease Stages

To study the virus-induced expression changes in host genes during disease progression, a panel of in situ assays have been developed over the years by different groups [3,24,25,36,56,70,71,92,114]. Newly improved assays, including the in situ hybridization for detection of CRPV DNA in CRPV-infected tumor tissues and improved RNA-ISH analysis to detect CRPV E4 transcripts have been applied to recent studies [82]. To validate host gene expression in the infected tissues, we have identified a panel of cancer-related genes that are upregulated in advanced CRPV lesions (Table 2, based on secondary analyses of our published RNAseq dataset) [24,25,82]. These include the biomarkers PCNA, Cyclin E, and MCM7 [25] which are also upregulated in HPV-associated cancer tissues (Table 2) [82,115,116,117,118,119,120,121,122]. A good example is pro-inflammatory molecule calcium-binding protein A9 or S100A9 which is highly dysregulated in both CRPV-infected tissues [82] and HPV-associated cancers [123,124,125,126]. These striking parallels between the rabbit model and the HPV cancers further enhance the value of this preclinical model for new targeted therapies. To facilitate study in different immune cell populations, we have also developed antibodies to rabbit T cell surface markers including a CD4 T cell antibody that has been used successfully for in vivo depletion studies [25,127]. These antibodies are useful for the validation of our earlier observations that fewer T cells (CD4 and CD8) infiltrate in tumors relative to those undergoing regression [41].

## 4. Rabbits for Studying Viral–Host Interactions during Tumor Progression

Rabbits have been used for studying a number of human diseases, including papillomavirus infections [128]. The host genetic background, including HLA class II alleles, plays an important role in HPV-associated disease progression and cancer development [129,130,131,132,133]. Similarly, rabbit MHCII has been linked to CRPV-induced tumor regression [53,54]. In agreement with these findings, we and others have demonstrated, using a variety of rabbit strains, that the host genetic constitution plays a role in disease outcome of CRPV infections [7,8,33,50,54,65,134]. Different responses to the same CRPV genomic construct have been reported in outbred and inbred rabbits in our studies [33,50]. We generated transgenic rabbits, including EJ-ras and HLA-A2.1 rabbits, to facilitate determination of the role of host oncogene and immune responses in the CRPV infection [33,46,50,134]. In recent years, novel gene modification technologies, especially CRISPR editing, have enabled rapid production of gene-modified rabbits [135,136,137].

### 4.1. Inbred and Outbred Rabbits

Most studies have used outbred rabbits that are supplied by several vendors including Charles River, Robinson, and Covance [3,50,61]. During the early years of our studies, we used rabbits from each of these suppliers [36,39,110,138]. While rabbits from different suppliers are all susceptible to CRPV infections, we did observe different natural regression rates following infections [39]. To maintain consistency from study to study, we have used the same supplier for most of our studies in the past two decades [8]. The inbred rabbit strain (EIII/JC) has been maintained in our facility for over thirty years and was originally acquired from NIH [33,50]. These inbred rabbits appear to be normal except for a heightened sensitivity to noise. These rabbits showed higher regression rates after CRPV infection [33,49]. In our previous studies, we have tested our CRPV mutant constructs on both outbred and inbred rabbits; the results are summarized in Table 3 [50]. It would be interesting to compare the host gene expression profiles after CRPV infections in these different rabbit strains.

### 4.2. Transgenic Rabbits

To understand the pathogenesis and tissue-tropism of CRPV in rabbits, we generated a CRPV/EJ-ras transgenic rabbit strain [48]. We observed that the tissue specificity of CRPV DNA expression in these rabbits was the same as in the virion-infected wild-type animals. It appears that the strict tissue-tropism of CRPV is controlled by the URR of the CRPV genome [46,47,48,139].

To facilitate vaccine development for HPV-associated diseases and cancers, we also developed an HLA-A2.1 transgenic rabbit model to test HPV vaccines in the context of a human MHCI background (HLA-A2.1) [134,140]. We have tested the immunogenicity of several known and unknown HLA-A2.1 restricted epitopes delivered by either DNA or peptides and have demonstrated both the prophylactic and therapeutic effects of these candidates [35,51,134,141]. The HLA-A2.1 transgenic rabbit model will continue contributing to future studies that lead to novel prophylactic and therapeutic strategies against HPV-associated diseases and cancers.

### 4.3. Novel Genetically Modified Rabbits

The recent development of gene-editing technologies has brought new tools to the development of animal models, especially for species for which germ-line embryonic stem cells (ESCs) are not available [55]. The attempts to produce gene-targeted rabbits date back two decades, after Chesne et al. reported the successful cloning of rabbits by somatic cell nuclear transfer [142]. The idea was to generate targeted mutations, for example, a gene knockout in the somatic cells (bypassing the need for ESCs), and to use these cells for animal cloning. Knockout pigs and cows had been produced via this strategy [143]. Unfortunately, despite large numbers of embryo transfers, no gene-targeted rabbits were cloned and produced using this approach.

The first gene knockout rabbit was produced shortly after zinc finger nuclease (ZFN) was introduced to researchers [144]. This first-generation gene editing nuclease (GEN) was quickly replaced by TALEN and then CRISPR/Cas9. To date, CRISPR/Cas9 represents the most commonly used GEN in the production of rabbit models [55]. Our group reported the first success in producing gene knockout rabbits by CRISPR/Cas9 in 2014 [135]. More than ten animal lines were efficiently produced, highlighting the power of CRISPR/Cas9 in the gene editing of rabbits. Later, in 2016, we reported that the efficiency of gene knock-in in rabbits by Cas9 or TALEN can be improved two–five-fold when a small molecular compound RS-1 is used [145].

In 2017, we reported the production of multiple lines of immunodeficient rabbits [138]. The targeted knockout genes include Foxn1, Il2rg, and Rag2. Foxn1 is essential for thymus and hair follicle epithelial cell development. The knockout of Foxn1 leads to the hairless “nude” phenotype and an impaired T cell development, as shown in athymic nude mice [146]. Il2rg is a gene that codes for the common gamma chain (γc), which is a cytokine receptor sub-unit that is common to the receptor complexes for different interleukin receptors. These include IL-2, IL-4, IL-7, IL-9, and IL-15. The loss-of-function mutation of Il2rg leads to defective B and T cell development and subsequently to severe combined immunodeficiency (SCID) disease [147]. Rag2 is involved in the V(D)J recombination process for B and T cells and is essential for the generation of mature B and T lymphocytes. Individuals with defective Rag2 therefore also often suffer from SCID. These immunodeficient rabbit lines will be useful for delineating the contributions of the B and T cells in viral pathogenesis, and for developing therapeutic strategies in the CRPV rabbit model.

## 5. Summary and Conclusions

HPV infection causes approximately 5% of human cancers and 30% of all cancers caused by infectious agents [148]. Most of the HPV infections (>90%) are cleared within two years because the host immune system is effective in eliminating HPV infections in most situations [148]. The CRPV rabbit model has provided opportunities to study the fine balance between viral oncogenicity and immunogenicity in deciding disease outcomes over the past several decades [1,3,8,149,150]. In addition to the key role of adaptive immune responses, we and others have also demonstrated that innate immune modulators, such as select cytokines, play a role in viral persistence and tumor progression [34,141]. However, an in-depth understanding of viral pathogenesis in the rabbit model has been delayed due to the slow advancement in whole genome sequencing and annotation of the rabbit genome [8]. Only recently were we able to obtain the genome-wide transcriptome profile of CRPV-infected lesions [83]. These datasets identified many parallel changes in different signal transduction pathways/genes that have been reported in HPV studies [150], which further confirmed the applicability of the CRPV rabbit model to HPV pathogenesis. The noticeable limitation for most human studies is that they have focused on HPV disease at a single time point assuring that the dynamics of tumor progression are difficult to follow [151]. This limitation can be overcome by using the rabbit model with predictable disease outcomes within a reasonable time frame. We can monitor dynamic changes at different disease stages and determine how the balance of oncogenicity and immunogenicity is associated with cancer development. Equipped with the availability of novel gene-modified rabbit lines, we expect to conduct more mechanistic studies leading to significant contributions to the deeper understanding of HPV pathogenesis.

The CRPV model continues to hold great promise for mechanistic studies of papillomavirus-associated disease progression or regression, especially with recent technological advances such as single-cell omics which provide unprecedented opportunities to analyze the complexities of biological systems at the single cell level. Novel hypotheses, including the dynamic changes in the balance of oncogenicity and immunogenicity during cancer development can be tested in this model system in future studies based on newly acquired knowledge as well as unique resources and reagents that will continue to be established for rabbit researchers.

## Figures and Tables

**Figure 1 viruses-14-01964-f001:**
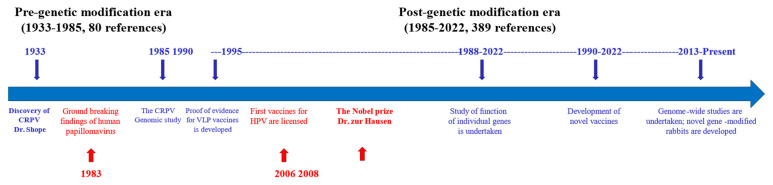
Several milestones of the CRPV rabbit model (blue) and HPV study (red). The rabbit model has played a pivotal role in HPV vaccine development and better understanding of HPV pathogenesis. The research using the rabbit model can be divided into two periods based on the first report on the genomic sequence of CRPV: pre-genetic modification era and post-genetic modification era. The notable research activities on the rabbit model have continued to reduce over the last two decades.

**Figure 2 viruses-14-01964-f002:**
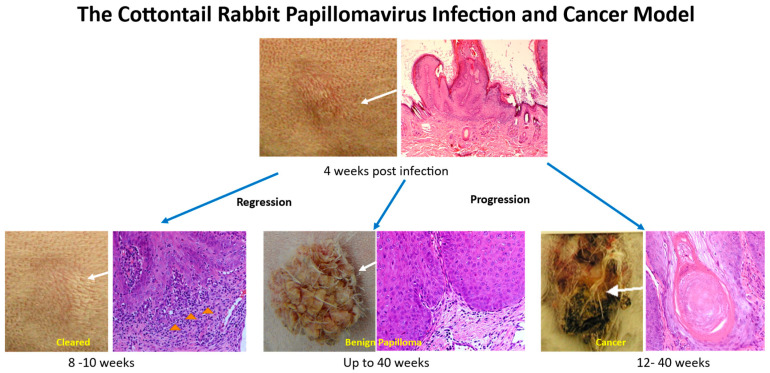
The CRPV rabbit model mimics HPV-associated infections and diseases with predictable disease outcomes at different time frames. Significant immune cell infiltrates (yellow arrows) were found in the tumors undergoing regression.

**Table 1 viruses-14-01964-t001:** Published mutant CRPV genomes with unique phenotypes.

Constructs (>300)	Tumor Phenotype	Cancer
Wild type (>3)	Latent, persistent, cancer	Yes, >12 months (Hu et al., 2002, 2005, 2009; Cladel et al., 2009, 2013)
Regressive strain (>5)	Regressive	No (Hu et al., 2002, 2005, 2009)
Hybrid, epitope etc., mutants (>200)	Varies	Maybe (Hu et al., 2002, 2005, 2009; Cladel et al., 2009, 2013; Bounds, 2010) and unpublished
E8 and SE6 mutants (>10)	Persistent, benign, and small	Maybe, >12 months (Hu et al., 2002, 2005, 2009; Cladel et al., 2009, 2013)
E7 mutant genomes (>5)	Persistent and benign	No (unpublished observations)
E6 and E7 codon optimized genomes (>20)	Regressive or Cancer	Yes, >3 months (Cladel et al., 2009, 2013)

**Table 2 viruses-14-01964-t002:** Representative molecules related to cancers and T cell functions that are significantly changed in both the CRPV-induced tumor tissues AND cervical cancer.

Genes	Changes in CRPV-Infected Tumors	Pathways
Krt1, 2, 3, 4, 7, 10, 14, 16, 78; Krt13, 75	UP/Down	Cytokeratin
KLF3, 10; KLF 1, 9, 11, 15	UP/Down	Keratinocyte proliferation
BRCA1, BRCA2, FANCD2, PCNA; DDR2	UP/Down	DNA damage
MAPK6, 13; MAPK12	UP/Down	p38 MAPKs
PCNA, CDK2, CASP8, ERBB3, PDCD5,6; TGFBR2, PDCD4	UP/Down	Cell growth and death
TP53I3, CDKN2A	UP	Tumor suppressor
CTLA-4, RNF149, Cblb, Rel, PD-L1; Gata3, NFATC1, 4, CD34, NR4A1, Foxp1, CD8b	UP/Down	T cell function
CXCL8, IFNgR1, STAT4; Cox-2, CX3CL1	UP/Down	Cytokines, chemokines, and ligands
IL1A, IL4R, IL10RA, IL13, IL17F, IL23A, IL36A, IL36g; IL6R, 11RA, IL13, IL16	UP/Down	Interleukins

**Table 3 viruses-14-01964-t003:** Rabbit strains used in our studies.

Rabbit Strain	Phenotype after Infection	References
Outbred	Persistent and cancer (wild-type CRPV) Regressive (regressive CRPV)	Hu et al., 2002, 2005, 2009; Cladel et al., 2009, 2013, 2019
EIII/JC inbred	Higher regression rate for wild-type CRPV	Hu et al., 2002, 2005, 2006, 2007, 2009
HLA-A2.1 outbred	Persistent and cancer (wild-type CRPV) with higher regression rates Regressive (regressive CRPV)	Hu et al., 2006, 2007, Bounds et al., 2009, Cladel et al., 2019

## Data Availability

Not applicable.

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
