# Peer review of "Modeling HPV-Associated Disease and Cancer Using the Cottontail Rabbit Papillomavirus"

_viruses, 2022, doi:10.3390/v14091964_

Round 1

Reviewer 1 Report

This is a well written and comprehensive review of the experimental studies on the Cotton Tail Rabbit Model (CRPV) of papillomavirus pathobiology and relevance for HPV associated disease.  I assume the authors were requested to focus on the studies originating from their group since there is little reference to other studies. 

I am surprised and disappointed that in this review there is no reference to the original papers from Shope especially since they reference Rous and Beard, discuss contribution to vaccines and infection consequences in domestic versus cotton tail rabbit species all of which deserve reference to the seminal papers from Shope

Shope, Richard E.; Hurst, E. Weston (31 October 1933). "Infectious Papillomatosis of Rabbits with a note on the histopathology"The Journal of Experimental Medicine58 (5): 607–624. doi:10.1084/jem.58.5.607PMC 2132321PMID 19870219

Author Response

This is a well written and comprehensive review of the experimental studies on the Cotton Tail Rabbit Model (CRPV) of papillomavirus pathobiology and relevance for HPV associated disease.  I assume the authors were requested to focus on the studies originating from their group since there is little reference to other studies. 

We thank the reviewer for these positive comments on the review. We also agree with comments concerning the narrow coverage of this review. In the revised version, we have expanded the content and included more findings from other laboratories to be more inclusive. However, the topics that we are focusing on in this review happen to be mostly from our recent work in the lab. We did not intend to exclude other labs’ contribution as our original assignment was to overview our research on the rabbit model. We published a comprehensive review back in 2019 [1] and don’t want to repeat the content in that review. We felt that it’s fitting to attract future scientists who are interested in this well-established but yet-to-be completely characterized model by emphasizing some exciting findings that are becoming lost and buried in the ever-increasing literature pool. As the reviewer might be aware of the small size of the current groups using this model (which kept shrinking over the past two decades), there remains an increased need from both NIH and industry for testing their novel anti-viral and immunotherapeutic compounds in this model. Our group has been a main contributor to such contract testing for the past several decades and our appreciation for the need to have effective preclinical models is “first hand”. With the advancements in new technologies, we hope we can further characterize this model system to make it more appropriate for HPV pathogenesis, and to pass on our experiences and findings to the next generation of scientists

I am surprised and disappointed that in this review there is no reference to the original papers from Shope especially since they reference Rous and Beard, discuss contribution to vaccines and infection consequences in domestic versus cotton tail rabbit species all of which deserve reference to the seminal papers from Shope

Shope, Richard E.; Hurst, E. Weston (31 October 1933). "Infectious Papillomatosis of Rabbits with a note on the histopathology". The Journal of Experimental Medicine58 (5): 607–624. doi:10.1084/jem.58.5.607. PMC 2132321. PMID 19870219

We apologize for excluding this primary publication of the CRPV model by Dr. Shope (and we are well familiar with much of these early publications). In the revised review that we have added and commented on his original discovery and the connection between our study with this early groundbreaking study.  For the past several decades, we have used this model system to test vaccines, novel therapeutic approaches but struggled with in depth understanding of this model especially at the molecular level.  Since we reported the first in depth genome-wide analysis on CRPV tumors, we realized that there are striking similarities between CRPV tumors and HPV associated cancer. We have included a new diagram to highlight the timelines of several key discoveries and their parallels with HPV research.

Reviewer 2 Report

A review article should be a comprehensive summary about a certain topic or in this case about an animal model system in a certain research area, describe recent major advances and discoveries, point out significant gaps in the research and current debates as well as suggestions of where research might go next.

In contrast, this review manuscript summarizes predominantly the findings of the authors (as demonstrated by the fact that “Christensen” has 49 of 143 citations in the reference list) and neglects important contributions from other groups. This is also supported by a statement of the authors in the summary in line 311 “The work of our laboratory is delineated in this review”. However, this is not the intention of a general review that should encompass all the work related to the title of the review, which implicates parallels between the mechanism of carcinogenesis of CRPV in the rabbit and HPV induced cancer in humans.

Furthermore, the authors tend to claim findings that have been described much earlier by others, e.g. the fact that pre-wounding of the skin increases the efficiency of infection in rabbits. This has been described already by W.F. Friedewald in the Journal of Experimental Medicine 1944 vol. 80: 65-82, by Peyton and Rous 1934 and already in the first description of infectious papillomatosos of rabbits by R.E. Shope 1933, J. Exp. Med 58, 607.

The same is true for the statement of the authors in line 108 “This is the first proof-of -evidence study to show that papillomavirus can be transmitted through the bloodstream and induce local infections at pre-wounded sites of domestic rabbits”. This experiment has been performed already by R.E. Shope in the above-mentioned paper 1933 – see page 615 at the bottom – and not as the authors claim for the first time in their publication in 2019. 

The content of the manuscript is not very well structured and confuse and the title is misleading. In light of the before mentioned points of criticism the authors need to rewrite the manuscript to summarize all the findings also from other groups with respect to parallels between the mechanism of carcinogenesis of CRPV in the rabbit and HPV induced cancer in humans.

Author Response

A review article should be a comprehensive summary about a certain topic or in this case about an animal model system in a certain research area, describe recent major advances and discoveries, point out significant gaps in the research and current debates as well as suggestions of where research might go next.

We thank the reviewer for pointing out the limitation of this review which focused on mostly on our studies in recent decades (as discussed above, this was the focus of our primary assignment). We had published a comprehensive review back in 2019 [1] and decided to focus on more recent and specific studies we had conducted. The rabbit papillomavirus model is currently used in only a very small number of laboratories in the world due to reduced interest after the post-vaccine era. However, we are keen to promote this valuable model, and are taking advantage of this opportunity to emphasize some recent advancements using modern genome-wide analysis to gain in-depth understanding of the molecular changes during infection using genetically modified viruses and animals.

In contrast, this review manuscript summarizes predominantly the findings of the authors (as demonstrated by the fact that “Christensen” has 49 of 143 citations in the reference list) and neglects important contributions from other groups. This is also supported by a statement of the authors in the summary in line 311 “The work of our laboratory is delineated in this review”. However, this is not the intention of a general review that should encompass all the work related to the title of the review, which implicates parallels between the mechanism of carcinogenesis of CRPV in the rabbit and HPV induced cancer in humans.

Again, we did cite a lot of our studies in this review which coincidently would be a similar listing of citation should a different author write a review on the CRPV rabbit model. To acknowledge the reviewer’s comments, we included additional work from other laboratories in the revision. We believe the revised version is more inclusive and comprehensive.

Furthermore, the authors tend to claim findings that have been described much earlier by others, e.g. the fact that pre-wounding of the skin increases the efficiency of infection in rabbits. This has been described already by[2] W.F. Friedewald in the Journal of Experimental Medicine 1944 vol. 80: 65-82, by Peyton and Rous 1934 and already in the first description of infectious papillomatosos of rabbits by [3, 4]R.E. Shope[5, 6] 1933, J. Exp. Med 58, 607.

Again, we thank the reviewer for listing these pioneer works which we included in our previous review. Dr. Shope’s elegant study published in JEM in 1933 described in detail how they generated the infectious materials and passaged infection in both wild and domestic rabbits. They did mention a scarification method used for inoculation with a large number of infectious virions in the rabbits. This method works only for infectious viruses which most laboratories no longer have stocks. The reviewer will appreciate that we could not generate infectious virus in vitro or in the domestic rabbits. Fortunately, the viral stock we used was generated in 1992 and is very stable at -80oC but it is a finite resource! A method to increase viral infectivity would be invaluable in the long run. To better understand function of each viral genes, we have developed a panel of mutant viral DNA cloned in a bacterial plasmid. The Shope scarification method did not work well for viral DNA infection in vivo. Our colleagues have tried many different delivery methods to inoculate viral DNA and none was optimal (we reviewed this in the revised version). Our lab utilized the turpentine/acetone method [7, 8]  and observed improvement in the outcome but this was very irritating for the research staff and treated animals[9]. In addition, hash chemicals to increase skin irritation are increasingly difficult to justify to research committees. We therefore developed this pre wounding strategy (scarify-3 days before inoculation-wound/inoculate virus or viral DNA) which has shown high consistency and reproducibility and uses no chemical pretreatments. Although we are not first to report about wounding during inoculation, the pre-wounding strategy is a revolutionary advancement for viral DNA infection (particularly mutant viral genomes with reduced infectivity) for the rabbit model and was further applied to the mouse papillomavirus model. We cited many studies from our lab because we happen to be the group that generated many of the findings emphasized in this review. We included additional work from other laboratories to be more inclusive and comprehensive.

The same is true for the statement of the authors in line 108 “This is the first proof-of -evidence study to show that papillomavirus can be transmitted through the bloodstream and induce local infections at pre-wounded sites of domestic rabbits”. This experiment has been performed already by R.E. Shope in the above-mentioned paper 1933 [5]– see page 615 at the bottom – and not as the authors claim for the first time in their publication in 2019. 

We thank the reviewer for noting this experiment. You are correct that Dr. Shope conducted a blood transmission study using the viral suspension (wart exact) in his paper. In our study, in addition to virions, we also tested viral DNA intravenously which was the first. We corrected the text to present these combined findings more accurately.

The content of the manuscript is not very well structured and confuse and the title is misleading. In light of the before mentioned points of criticism the authors need to rewrite the manuscript to summarize all the findings also from other groups with respect to parallels between the mechanism of carcinogenesis of CRPV in the rabbit and HPV induced cancer in humans.

We acknowledge that the reviewer had difficulty following the structure of this review,  and have revised accordingly to include the reviewer’s suggestions. We also thank the review for these comments that help to greatly improve the content. We have revised the manuscript extensively by grouping relevant information together and added additional background/justification to improve the flow of the paper. The revised version includes more insights and implications of the findings on papillomaviruses in general and also aligns the mechanisms of carcinogenesis of CRPV with HPV associated cancers.

  1. Cladel, N. M.; Peng, X.; Christensen, N.; Hu, J., The rabbit papillomavirus model: a valuable tool to study viral-host interactions. Philos Trans R Soc Lond B Biol Sci 2019, 374, (1773), 20180294.
  2. Friedewald, W. F.; Rous, P., The initiating and promoting elements in tumor production; an analysis of the effects of tar, benzpyrene, and methylcholanthrene on rabbit skin. J. Exp. Med 1944, 80, 101-126.
  3. Beard, J. W.; Rous, P., A virus-induced mammalian growth with characters of a tumor (the Shope rabbit papilloma): II. Experimental alterations of the growth on the skin: morphological considerations: the phenomenon of retrogression. J. Exp. Med 1934, 60, 723-740.
  4. Rous, P.; Beard, J. W., A virus-induced mammalian growth with characters of a tumor (the Shope rabbit papilloma): I. The growth on implantation within favorable hosts. J. Exp. Med 1934, 60, 701-722.
  5. Shope, R. E.; Hurst, E. W., Infectious papillomatosis of rabbits; With a note on the histopathology. J. Exp. Med 1933, 58, 607-624.
  6. Escudero Duch, C.; Williams, R. A.; Timm, R. M.; Perez-Tris, J.; Benitez, L., A Century of Shope Papillomavirus in Museum Rabbit Specimens. PLoS One 2015, 10, (7), e0132172.
  7. Kreider, J. W.; Cladel, N. M.; Patrick, S. D.; Welsh, P. A.; DiAngelo, S. L.; Bower, J. M.; Christensen, N. D., High efficiency induction of papillomas in vivo using recombinant cottontail rabbit papillomavirus DNA. J Virol Methods 1995, 55, (2), 233-44.
  8. Friedewald, W. F.; Rous, P., The determining influence of tar, benzpyrene, and methylcholanthrene on the character of the benign tumors induced therewith in rabbit skin. J. Exp. Med 1944, 80, 127-144.
  9. Cladel, N. M.; Hu, J.; Balogh, K.; Mejia, A.; Christensen, N. D., Wounding prior to challenge substantially improves infectivity of cottontail rabbit papillomavirus and allows for standardization of infection. J. Virol. Methods 2008, 148, (1-2), 34-39.

Round 2

Reviewer 2 Report

The authors hace improved the manuscript, but still they have to mention in the introduction last paragraph in their sentence "In the current
review, we will focus on some important findings, mostly from our laboratory" as they still ignore a lot of literature related to viral pathogenicity of CRPV as for example papers showing carcinogenic activity of CRPV E2 and other findings. If a newcomer to the field reads this review she/he will not be directed to other literature which is published and believe that this review is a comprehensive one.

Author Response

The authors hace improved the manuscript, but still they have to mention in the introduction last paragraph in their sentence "In the current
review, we will focus on some important findings, mostly from our laboratory" as they still ignore a lot of literature related to viral pathogenicity of CRPV as for example papers showing carcinogenic activity of CRPV E2 and other findings. If a newcomer to the field reads this review she/he will not be directed to other literature which is published and believe that this review is a comprehensive one.

We are sorry that the reviewer must have an incorrect version as the sentence "mostly from our laboratory" was already eliminated from the revised version (viruses-1846585).